# A Green Sintering-Free Binder Material with High-Volumetric Steel Slag Dosage for Mine Backfill

**Bolin Xiao** [1,*] **, Huatao Huang** [2] **and Jingyu Zhang** [3]

1. State Key Laboratory of High-Efficient Mining and Safety of Metal Mine Ministry of Education, University of Science and Technology Beijing, Beijing 100083, China
2. Yunnan JCHX Mining Management Co., Ltd., Kunming 650217, China
3. Xinjiang Daming Mining Group Co., Ltd., Hami 839121, China
* Correspondence: bxiao@ustb.edu.cn; Tel.: +86-132-6950-0905

**Abstract:** Cemented paste backfill (CPB) is a sustainable mining method that has been increasingly utilized. Demand for high-performance and low-cost binder material is one of the limitations in CPB utilization. This work aims to examine a new, green, and economical steel-slag-based binder for CPB and to explore valorization techniques of steel slag (SS). Proportioning experiments were performed to obtain the best binder recipes for various steel slag dosages. The hydration heat, hydration products, and pore structure of a high-volumetric steel slag binder (H-SSB) were further inspected. Results show that the H-SSB, which contains 50 wt.% of SS, has a competitive strength performance superior to ordinary Portland cement (OPC) regardless of its 30–50% lower cost than OPC. The 7-day and 28-day strengths of H-SSB CPB are 1.24 and 0.74 MPa, respectively, which meets the meets of most free-standing backfill applications. The H-SSB generates less hydration heat and a larger amount of gel and ettringite hydrates than OPC in its early hydration, which can reduce the thermal expansion risks and strengthen the mechanical properties of CPB. Though the H-SSB CPB has a larger porosity than OPC-CPB at 28-day curing (45% vs. 37%), most pores are small and uniform in diameter (500–2000 nm), which is less harmful to CPB strength development. The H-SSB has secondary hydration effects in the long-term age, which can fill and refine the pore structure. The proposed H-SSB has benefits in reducing backfill costs, minimizing green gas emissions, and extending steel slag valorization techniques that can promote sustainable development of the mining and steel industries.

**Keywords:** cemented paste backfill; binder material; steel slag; hydration; green mining

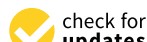



## 1. Introduction

Mining activities provide sufficient raw materials for civilization while leaving an environmental footprint which includes solid waste discharge, water pollution, and ground subsidence [1]. Eco-friendly and sustainable mining techniques are attracting noticeable investigations from all over the world. Cemented paste backfill (CPB) is one such widely used mining method [2]. CPB is generally a mixture of binder materials, tailings, water, and additives used to fill excavated voids [3]. Thus, mining by-products, mainly tailings, are recycled; other benefits of effective ground support, safe working platform, and the capability to achieve maximum ore recovery rate [4].

One of the limitations in CPB utilization is the relatively high cost when compared with other mining methods. Underground paste backfilling contributes up to 20% of the overall cost structure of an underground operation [5]. Specifically, the binder material can make up 40–75% of the total backfill cost, depending on the strength requirement of the filling body, physical and chemical compositions of tailings, and characteristics of backfill plant systems [6]. Typically, ordinary Portland cement (OPC) is used as the binder material, which accounts for 4–20% of the CPB's mass fraction [7,8]. OPC is generally partially

replaced by pozzolanic materials such as metallurgical slag, fly ash, zeolite, and sintered soil to reduce the cost [9–12]. Many other researchers have developed customized backfill binders to pursue lower cost, high performance, or value-added utilization. For example, S. Tuylu investigated the effect of using different fly ash and chemical structures on the mechanical properties in CPB [13]. M. Petlovanyi assessed the expediency of applying the binder material mechanical activation in a cemented rockfill [14]. H. Jiang et al. tested the performance of alkali-activated slag-based CPB considering the coupled effects of activator composition and temperature [15]. B. Ercikdi et al. used granulated marble wastes and waste bricks as mineral admixtures in CPB with sulphide-rich tailings [16]. A. Wu et al. investigated hemihydrate phosphogypsum as cementitious material or aggregate in CPB, and effective utilization strategies were discussed in an application case [17]. J. Qiu et al. produced an eco-friendly binder using calcined steel slag, metakaolin, and silica fume [18]. B. Xiao et al. also presented a steel slag binder (SSB) without thermal treatment for backfill, where the hydration, strength, pore structure, and cost of the SSB-CPB were analyzed [19]. Note that cementitious materials made with industrial solid waste have prominent advantages of low cost and efficient utilization.

Ground granulated blast furnace slag (GGBFS) is a by-product of iron manufacturing which has been widely utilized in cement and concrete industries worldwide. The great demand for GGBFS makes it a precious resource with a surging price close to OPC. Steel slag (SS), another by-product of the steel-making industry, is a common industrial solid waste that has not yet been effectively managed in China. More than 100 million tons of steel slag are generated annually, accounting for more than half of the world's annual total output [20]. The utilization rate is only 29.5%, causing environmental problems [21]. Steel slag has similar mineral compositions to clinker, which makes it a potential supplementary cementitious material (SCM) for cement production. The presence of expansive compounds, low binding property, and hardness are key barriers in its SCM applications [22]. Reviews on steel slag applications in cement and concrete showed that steel slag has conditionally positive or negative effects such as improving workability, decreasing hydration heat, prolonging the setting time, and reducing the mechanical properties; the steel slag incorporation rate in cement has a recognized level of 10–35 wt.% [22–25]. Slags could be utilized together with flue gas desulfurized gypsum (FGDG), which is an industrial by-product obtained by using lime-limestone to recover sulfur dioxide in coal-fired flue gas. The applications of FGDG in cement, asphalt, motor, grout, and concrete have different benefits and low environmental risks [26]. Therefore, the utilization of steel slag is still limited; all kinds of efforts in dealing with steel slag should be credited and of significance for sustainable development.

The increasing demand for cementitious material in mine backfill offers an ideal opportunity for steel slag treatment. Unfortunately, the existing studies on massive steel slag utilization in CPB are limited and insufficient. This work presents a green sintering-free binder with a high-volumetric steel slag dosage for mine backfill. The binder hydration and the CPB strength properties are examined. The findings provide a new alternative binder for reducing backfill cost and a new approach for steel slag valorization.

## 2. Materials and Methods

### 2.1. Materials and Sample Preparation

#### 2.1.1. Steel Slag (SS)

The SS, a basic oxygen furnace slag, was obtained from Fujian Sanming Iron and Steel Group in Sanming city of south China. The SS was magnetically separated to recycle the ferrous metals before crushing and grinding with an industrial ball mill pulverizing system, where SS was ground into a specific area (SA) of 430–460 $m^2$/kg. The SA is slightly higher than OPC, which is generally 400 $m^2$/kg. The industrial experience of slag SA is in the range of 400–500 $m^2$/kg, considering its grinding cost and reactivity [27]. The $f$-CaO content in the SS powder is less than 8%. The Mason basicity of the SS was 2.18, indicating good hydration potential. The chemical compositions of the SS are given in Table 1.

**Table 1.** Main chemical compositions of the materials.

| Item | TFe | SiO₂ | Al₂O₃ | CaO | MgO | SO₃ | Others |
|---|---|---|---|---|---|---|---|
| SS (%) | 16.6 | 18.9 | 4.5 | 46.2 | 4.6 | 0.3 | 8.9 |
| GGBFS (%) | 0.7 | 30.7 | 14.0 | 43.2 | 7.4 | 0.2 | 3.8 |
| FGDG (%) | 0.2 | 1.8 | 0.4 | 50.2 | 0.1 | 47.1 | 0.3 |
| Tailings (%) | 10.3 | 33.1 | 3.7 | 26.1 | 4.4 | 0.1 | 22.3 |

### 2.1.2. Ground Granulated Blast Furnace Slag (GGBFS)

The granulated blast furnace slag, an iron blast furnace by-product, was also obtained from Fujian Sanming Iron and Steel Group (Sanming, China). It mainly consists of silicate and aluminosilicate of melted calcium, which exhibits hydration activity through alkali activation [28]. The slag was ball-milled as an ultra-fine powder, namely GGBFS, whose specific area was 450–480 m²/kg. The chemical compositions of the GGBFS are given in Table 1.

### 2.1.3. Flue Gas Desulfurized Gypsum (FGDG)

The main component of FGDG is calcium sulfate dihydrate, which can replace natural gypsum as an activator or set retarder in cement and production. The FGDG was sampled from a thermal power plant and used as a sulfate activator in the binder. The FGDG was dried at 40 °C in an oven to remove the free water and then sieved through a 45 μm mesh screen. The chemical compositions of the FGDG are given in Table 1.

### 2.1.4. Tailings

Natural iron ore tailing was used as aggregate, which was obtained from a mineral processing plant in Fujian Province, China. Thickened tailings was dried at 60 °C in a dry chamber to remove the free water completely. Then, the dry agglomeration was destroyed with a pestle. The grain size distribution of the tailings was analyzed through Mastersizer 2000 laser diffraction (Malvern, UK), and the result is shown in Figure 1. It can be seen that the percentage of particles finer than 20 μm is 39.6%, and the tailings could be classified as medium tailings. The particles finer than 75 μm are 77.4 wt.%. Some chemical and physical properties of the tailings are shown in Tables 1 and 2.

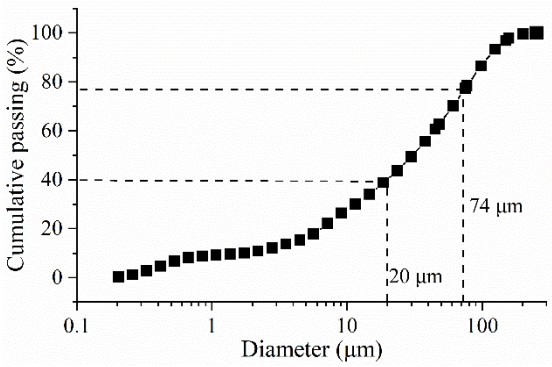

**Figure 1.** Grain size distribution of the tailings.

**Table 2.** Physical properties of the tailings.

| Item | G_s | θ (°) | P (%) | D₁₀ (μm) | D₃₀ (μm) | D₅₀ (μm) | D₆₀ (μm) | D₉₀ (μm) | C_u | C_c |
|---|---|---|---|---|---|---|---|---|---|---|
| Tailings | 2.98 | 41.6 | 33.8 | 1.83 | 11.66 | 30.01 | 44.96 | 118.3 | 24.56 | 1.65 |

$G_s$ = specific gravity; $\theta$ = repose angle; P = porosity; $C_u$ = coefficient of uniformity; $C_c$ = coefficient of curvature; $D_x$ = featured diameter corresponding to x% of the particles passing the featured sieves.

*2.2. Sample Preparation*

2.2.1. CPB Samples

The green binder is made of SS, FGDG, and GGBFS, which are all industrial wastes. All these three raw materials are pulverized and mixed without sintering. The SS dosage in cement is 10–30 wt.%, as mentioned in the introduction. In order to explore the influence, maximum, and best dosage of SS, experiments on five SS dosage levels were designed in Table 3. A Chinese standard of 42.5# OPC was used as a control. The binder to tailings ratio is set to 1:4 following the practical configuration; it is a common value in most mine sites in China [29,30]. The best five binder recipes were determined from the corresponding CPB 28-day strength. To prepare the CPB samples, all the required amounts of SS, FGDG, GGBFS, tailings, and water were mixed for 5 min in a mixing machine to reach a homogeneous state. The fresh slurry was poured into a triple cube mold with a side length of 7.07 cm. The samples were wrapped with a plastic film to avoid water evaporation and then cured in a stable temperature-controlled chamber of 20 ± 0.5 °C. There were a totally of 162 CPB samples prepared.

**Table 3.** Mix proportions of the CPB.

| Binder Recipe (A + B + C = 100%) | | | Binder to Tailings Ratio | Solid Mass Concentration, % | Curing Period, Days | Best Formula (SS/FGDG/GGBFS) |
|---|---|---|---|---|---|---|
| SS wt.%, A | FGDG wt.%, B | GGBFS wt.%, C (Rest) | | | | |
| 35 | 16, 18, 20, 22, 24 | 49, 47, 45, 43, 41 | 1:4 | 64 | 7, 28 | 35/24/41 |
| 40 | 16, 18, 20, 22, 24 | 44, 42, 40, 38, 36 | 1:4 | 64 | 7, 28 | 40/16/44 |
| 45 | 16, 18, 20, 22, 24 | 39, 37, 35, 33, 31 | 1:4 | 64 | 7, 28 | 45/20/35 |
| 50 | 16, 18, 20, 22, 24 | 34, 32, 30, 28, 26 | 1:4 | 64 | 7, 28 | 50/20/30 (H-SSB) |
| 55 | 16, 18, 20, 22, 24 | 29, 27, 25, 23, 21 | 1:4 | 64 | 7, 28 | 55/16/29 |
| OPC 100% (control) | | | 1:4 | 64 | 7, 28 | - |

2.2.2. Paste Samples

After the best binder recipes on five SS dosage levels were obtained, namely the ratio of SS/FGDG/GGBFS, their paste samples were prepared to identify the hydration properties. The paste sample was a mixture of binder and water with a ratio of 1.0 to mimic the high water content of CPB. The mixing, sealing, and curing procedures were the same as the CPB samples. After maintaining the curing period, the paste samples were oven-dried at 40 °C until the mass difference was less than 0.1% within 4 h. Then the dried sample was fractured and ground into powder using a pestle and mortar.

*2.3. Experimental Program*

2.3.1. Unconfined Compression Strength (UCS)

Although the CPB mechanical strength is affected by loading conditions and many other factors [31], the unconfined compression strength (UCS) is the most commonly used parameter to evaluate the binder and CPB performance. The test was conducted using an HYE-300 (SANYU, Cangzhou, China) compression machine following the ASTM C109 standard. The cube CPB sample was compressed on the lateral side at a deformation rate of 1 mm/min controlled by the electronic universal loading system. The strain and stress were automatically recorded from the in-built LVDT and pressure sensors. Three cube samples were tested for each mixing proportion to ensure accuracy and repeatability, and the mean value was reported as the USC of the sample.

The experimental programs are illustrated in Figure 2.

2.3.2. Heat of Hydration (HoH)

The HoH is the heat liberated from the reaction between binder and water. Monitoring the HoH gives the binder material detailed hydration properties, particularly kinetics. The binder (H-SSB and OPC) HoH was monitored using a TAM Air 8-channel isothermal

calorimeter (TA Instruments, New Castle, PA, USA) following the ASTM C1702 standard. The binder HoH monitoring lasted for three days at an operating temperature of 20 °C and a water/cement ratio of 0.35, which were common values in hydration heat tests.

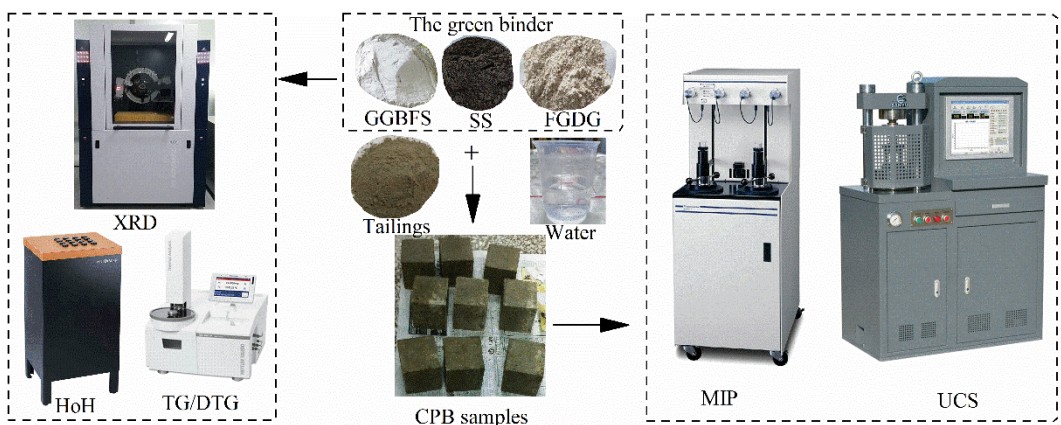

**Figure 2.** Brief diagram of the experimental programs.

### 2.3.3. X-ray Diffraction (XRD)

The XRD analysis was used to inspect the crystalline mineral components of the hydration products of the binder material. XRD was performed on the binder paste powder samples, where corundum was added as an internal standard in a mass ratio of 1:1. A D8 Advance multi-purpose powder diffractometer (Bruker, Billerica, MA, USA) was used. The scanning was carried out at a 2θ range of 10° to 90° with Cu Kα radiation at a voltage of 40 kV and current of 40 mA. The scanning step width and speed were 0.02° and 1°/min, respectively.

### 2.3.4. Thermogravimetry (TG/DTG)

TG/DTG was performed on paste samples to study the thermal properties of the hydration products. The composition information can be deduced from the mass and heat flow curves since minerals have distinct thermal decomposition characteristics. TG/DTG analysis was carried out using a TGA/DSC3+ thermal analyzer (Mettler Toledo, Switzerland). About 10 mg of paste powder was heated in a nitrogen atmosphere from room temperature to 1000 °C at a heating ramp of 10 °C per minute. The apparatus can record the change in weight loss, heat flow, and transition during the test.

### 2.3.5. Mercury Intrusion Porosimetry (MIP)

MIP is a pore size measurement technique that uses non-wetting liquid penetration to measure the size and volume of pores in porous solids. The CPB sample pore structure was inspected using AutoPore IV 9500 mercury porosimeter (Micromeritics, Norcross, GA, USA), which has a maximum pressure of 33,000 PSI and is capable of measuring a pore size range from 800 μm to 5 nm.

## 3. Results and Discussion

### 3.1. Strength Properties of the CPB with Green Binder Materials

There are five mixing proportions for a specific SS dosage of the green binder. The relations between the highest CPB strength and SS levels are shown in Figure 3.

The best recipes of the green binder (SS/FGDG/GGBFS), determined from the highest 28-day strength of the corresponding CPB and illustrated in Figure 3, are 35/24/41, 40/16/44, 45/20/35, 50/20/30, and 55/16/29, respectively. It can be seen that the amount of FGDG varies in the recipes. It can be ascribed to the complex hydration mechanisms in the green binder, which will be discussed in Section 3.2.

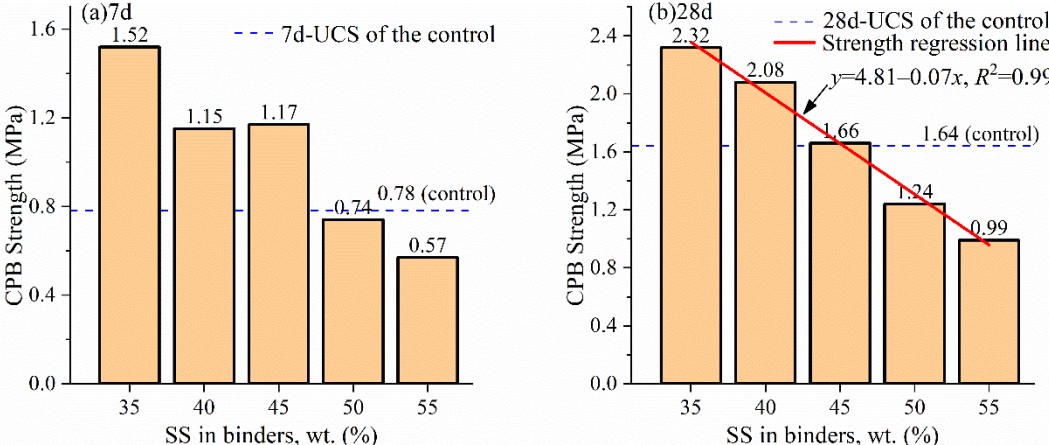

**Figure 3.** The highest CPB strength vs. SS dosage in the green binder at (**a**) 7 d curing and (**b**) 28 d curing.

The 28-day UCS of the CPB declines linearly from 2.32 MPa to 0.99 MPa as the SS dosage rises from 35 wt.% to 55 wt.%. The negative linear relation is regressed in the following Equation (1).

$$y = 4.81 - 0.07x, R^2 = 0.99 \tag{1}$$

where $y$ is the 28 d UCS of the CPB, and $x$ is the mass fraction of SS in the green binder.

However, the negative linear correlation is not significant for 7 d strength, which may be caused again by the complex hydration process. Zhuang et al. showed that steel slag inhibits the early-age hydration of cement by inhibiting the precipitation of CH and C-S-H [32]. The reaction products are often metastable when incorporated with SS, leading to micro-structural changes at early ages [33]. The dominant reactions at early-age are changeable, which leads to variations in the early-age CPB strength.

The green binder has a good performance when compared to OPC. It can be seen from Figure 3 that CPB with 45 wt. % SS in the green binder has a roughly equal 28-day strength to the control OPC-CPB (1.64 MPa). Furthermore, its 7-day strength (1.17 MPa) is 1.5 times higher than the control (0.78 MPa). When a high-volumetric (50 wt.%) SS is used in the green binder (H-SSB), the 28-day CPB strength is 1.24 MPa, which is slightly lower than the OPC-CPB control; however, the 7-day CPB strengths are basically the same (0.74 vs. 0.78 MPa). The green H-SSB has a competitive strength performance regardless of its benefits of using solid waste raw material and a sintering-free process. From previous work, the estimated cost of the H-SSB can be 30–50% lower than OPC in China, considering the processes of manufacturing, transfer, and shipping [19].

The H-SSB has an inspiring, practical application prospect, considering that most free-standing backfill applications require a 28-day CPB strength of 1–2 MPa [6,34,35]. Moreover, the volumetric expansion issues are generally of concern in backfill [36]. The expansion problem is expected to be avoided since the content of $f$-CaO in the SS is 8%, which can be fully consumed by slag hydration. Therefore, it is worth inspecting the hydration properties and the H-SSB microstructure in the following sections.

### 3.2. Hydration Heat of the Green H-SSB

Figure 4 shows the hydration heat evolution of the H-SSB within 72 h. Overall, the H-SSB has a similar hydration heat curve as OPC, which can be divided into five stages initial mixing reaction, dormancy, strength acceleration, speed reduction, and steady development [37]. However, the H-SSB has low hydration heat generation than the OPC (78 J/G vs. 113 J/G), which can reduce the thermal stress in the CPB bulk. This is beneficial for engineering practice since previous studies showed that the mass hydration heat could change the mechanical behavior and lead to thermal cracking [38]. The lower cumulative hydration heat can be ascribed to two reasons. First, the H-SSB has 50 wt.% of SS, which is less than the clinker content in OPC. The main mineral composition in SS is $\beta$-$C_2S$,

which has much lower reactivity than clinker. Second, the H-SSB has 30 wt.% of GGBFS, which can consume parts of the SS hydration heat. The GGBFS can be alkali-activated and thermal-activated [39].

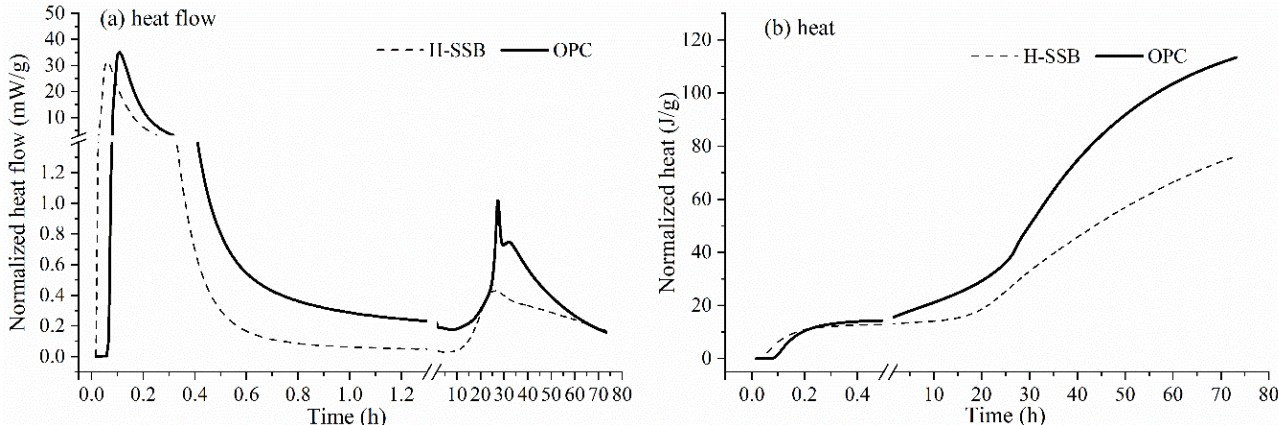

**Figure 4.** Hydration heat evolution of the H-SSB and OPC at 20 °C; (**a**) hydration heat flow within 72 h; (**b**) cumulative hydration heat.

There are two heat peaks during the hydration. The first peak occurs in the initial mixing phase (0–1 h), where the H-SSB has an earlier, nearly immediately peak generation than OPC. The first normalized heat peak value for H-SBB is 31.6 mW/g, which accounts for 90% of the OPC (35.1 mV/g). The peak is caused by the dissolution of FGDG and C3S, and the H-SSB has a higher FGDG content than OPC. The FGDG contains some un-reacted calcium oxide and sodium hydroxide, which will release heat after dissolving with water. The dissolution of FGDG releases faster heat than C3S, while the OPC contains more C3S, yielding a higher heat peak. The second main exothermic peak happens approximately 10 h after blending with water, where the OPC has 3.4 times higher peak value than the H-SSB (1.019 mW/g vs. 0.430 mW/g). Although many studies found that SS can prolong the dormant period and retard the setting time [32,40], the H-SSB does not exhibit the delay. The reason is that the H-SSB has coupled alkali and sulfate activating mechanism [20], which compensates for the low reactivity of SS. Furthermore, the free CaO and MgO, which has a low hydration rate, are fully consumed by the GGBFS, improving the H-SSB hydration. Another reason might be that the SS and GGBFS have a finer specific area of 430–480 m$^2$/kg than OPC, which a review showed that the slag hydration rate could be accelerated by increasing its fineness [41].

### 3.3. Hydration Properties of the Green H-SSB

3.3.1. Qualitative Analysis of the Hydration Products

The hydration products of the H-SSB and OPC were examined through XRD, and the results are stated in Figure 5. The main crystal hydration products of the H-SSB are ettringite and gypsum. Hydration of SS generates calcium hydroxide (CH). Then, the GGBFS, whose main composition is a solid solution of CaO-SiO$_2$-Al$_2$O$_3$, can react with the CH to produce calcium silicate hydrate (C-S-H) and calcium aluminate hydrate (C-A-H). Further reactions between C-A-H and sulfate produce the ettringite. These main reactions in H-SSB are in accordance with the XRD results, where CH peaks are absent, and ettringite peaks are found in H-SSB. Comparatively, the hydration products in OPC have multiple crystal minerals such as gypsum, hydrocalumite, calcium carbonate, and dolomite. The clinker hydration produces a large quantity of C-S-H and CH. The presence of calcium carbonate comes from CH carbonation [42]; then, parts of the calcium ions can be replaced by magnesium ions precipitating dolomite.

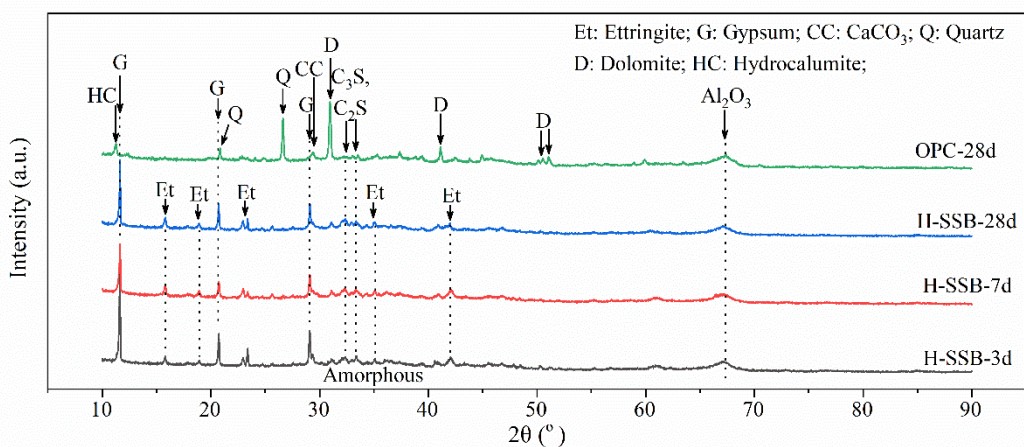

**Figure 5.** The XRD patterns of the H-SSB and OPC hydration products.

Additionally, the $C_2S$ and $C_3S$ peaks in the H-SSB are higher than OPC, indicating more un-reacted SS in the H-SSB. Un-reacted GGBFS is also found in the irregular bumps between 2θ degrees of 30 and 35°. These remaining particles are generally sealed in the CPB pore structure, which has positive secondary hydration effects of refining the pore structure and strengthening the mechanical properties [43].

### 3.3.2. Quantitative Analysis of the Hydration Products

The XRD patterns are uniform for the H-SSB 3 d, 7 d, and 28 d hydration. The hydration properties were further quantitatively inspected through TG/DTG analysis, shown in Figure 6.

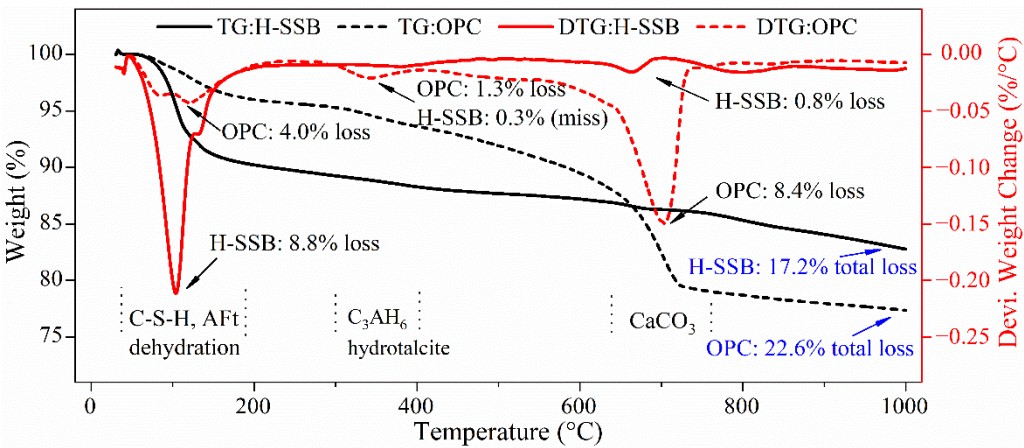

**Figure 6.** The TG/DTG curves of the H-SSB and OPC paste samples at 28-day curing.

The TG/DTG curve of the H-SSB is significantly different from the OPC. Overall, OPC has a higher total mass loss than H-SSB (22.6% vs. 17.2%), which signifies that there are more hydration products generated in OPC than H-SSB in 28-day curing. This explains the higher 28-day UCS of OPC-CPB in the previous Figure 3. However, the H-SSB has a considerably higher weight change peak than OPC at around 100 °C (8.8% vs. 4.0%), which is caused by dehydration of C-S-H and ettringite [44]. The more amorphous gel and ettringite hydrates are generated in the H-SSB through the aforementioned alkali, thermal, and sulfate-coupled activation. These gel and ettringite hydrates are key bonding agents for the early-age CPB [45]. Thus, the reason for the higher 7-day UCS of the H-SSB CPB in Figure 3 is interpreted.

Nevertheless, the OPC has an outstanding weight loss peak than H-SSB at around 700 °C (8.4% vs. 0.8%), which is caused by the decomposition of calcium carbonate. The

carbonation effects in the OPC contribute to the later-age CPB strength, compensating for the smaller amount of gel hydrates. Carbonation, which mainly refers to the reaction between CH and carbon dioxide, is hardly seen in the later-age H-SSB due to the absence of CH in its hydration products.

### 3.4. Pore Structure of the CPB at 28-Day Curing

The comparison of pore structures of the CPBs prepared by H-SSB and OPC cementitious materials was studied through MIP. Table 4 lists the pore size characteristics in the MIP test, and Figure 7 illustrates the pore structure distributions.

**Table 4.** The 28-day CPB pore size characteristics in the MIP test.

| Item | H-SSB CPB | OPC CPB |
|---|---|---|
| Total intrusion volume, mL/g | 0.3261 | 0.2831 |
| Total pore area, $m^2$/g | 5.118 | 0.53 |
| Median pore diameter (volume, V), nm | 1090.26 | 2687.86 |
| Median pore diameter (area, A), nm | 57.33 | 1947.38 |
| Average pore diameter (4 V/A), nm | 254.84 | 2135.23 |
| Porosity, % | 45.03 | 37.19 |

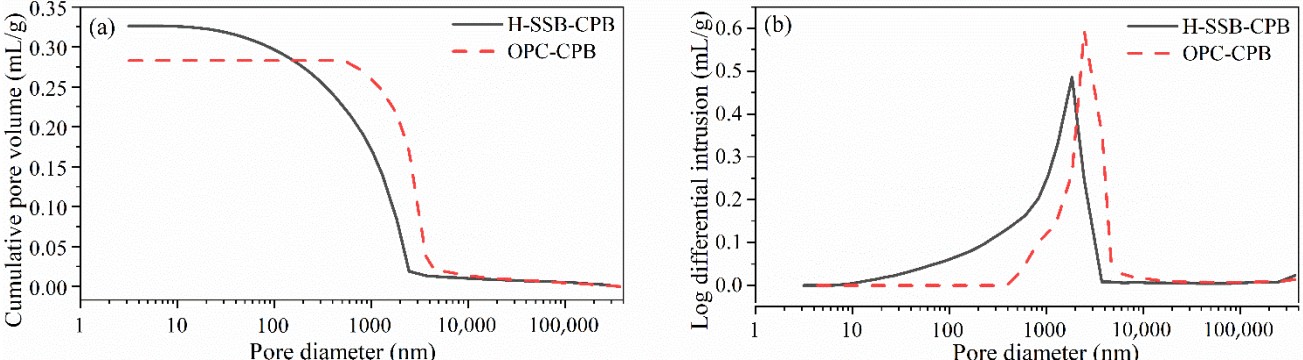

**Figure 7.** The MIP test results of the H-SSB and OPC CPB samples at 28-day curing: (**a**) cumulative pore volume distribution and (**b**) pore volume vs. pore size distribution.

The 28-day CPB porosity with H-SSB is 45.03%, which is larger than that of OPC (37.19%). Similar results were found in another study where CPB was incorporated with SS [46]. The greater porosity can be ascribed to the high specific gravity and low reactivity of the SS, which forms low-quality gel hydrates in the CPB pore structure [22]. It should be noted that the CPB porosity differences between OPC and SSB will be largely reduced in long-term curing [47], where the participated SS particles hydrate to fill and refine the pore structure [48].

Regardless of the greater porosity, the median pore diameter and average pore diameter (4 V/A) of the H-SSB CPB are 1090 nm and 254.84 nm, which are much smaller than OPC-CPB of 2687.86 nm and 2135.23 nm. It can be inferred that most pores in the H-SSB CPB are uniform pores with small diameters. From Figure 7, the H-SSB CPB has a wider pore distribution zone located in the range of 10–4000 nm, and more than 50% of pores are distributed in 500–2000 nm. Most pores in OPC-CPB are scattered in 500–6000 nm, and more than 80% are concentrated in the 1000–5000 nm range. From the IUPAC method, pores can be classified into micro-pores (<2 nm), meso-pores (2–50 nm), and macro-pores (>50 nm) [49]. The data of micro-pores below 2 nm cannot be obtained as the minimum pore size resolution of the MIP instrument is 3.1 nm. The H-SSB CPB has 5.57% meso-pores, the rest being macro-pores greater than 50 nm, while the OPC-CPB are all macro-porous.

The pore structure has a close relationship with the CPB mechanical behaviour. For example, a study showed that the pore size influenced the non-linear creep compression

behaviours of waste-rock backfill [50]. Another paper found that the proportion of macro-pores was inversely proportional to the UCS, while the proportion of micro-pores was proportional to the UCS [51]. Research also indicated that the permeability was dominated by the network of macro-pores, which constituted only ~4% of the total porosity/moisture content of the CPB [52]. The CPB pore structure is critical for a more thorough understanding of the functionality as a support structure during mining operations.

### 4. Conclusions

This work presents a novel, green, and sintering-free binder with high-volumetric steel slag for cemented pasted backfill, and the following conclusions can be obtained.

The 28-day CPB strength with the green binder is negatively proportional to the steel slag dosage from 35 to 55 wt.%. The green binder with 45 wt.% SS has competitive binding performance superior to OPC. A promising green H-SSB, whose recipe of SS/FGDG/GGBFS is 50/20/30 and has a 30–50% lower cost than OPC, can be a perfect alternative for CPB. The H-SSB CPB meets the comment strength requirements of CPB regardless of its other benefits of using solid waste raw material and a sintering-free process.

The H-SSB has similar hydration heat patterns as OPC, while its total hydration heat is much lower. The setting time is not prolonged due to its complex alkali, thermal, and sulfate activation mechanism. A large quantity of gel-like and ettringite hydrates are produced in its early hydration, making it a better early-age binding ability than OPC. These characteristics are beneficial to reducing CPB thermal expansion risks and strengthening the CPB mechanical properties.

Though the H-SSB CPB has a larger porosity than OPC-CPB at 28-day curing, most of its pores are uniform, small pores with narrow diameters ranging from 500–2000 nm. The smaller average pore diameter is less harmful to CPB strength development. The H-SSB has secondary hydration effects in the long-term age, which can fill and refine the pore structure, obtaining strengthened CPB mechanical properties.

Regardless of its good mechanical and hydration properties, the H-SSB has 50 wt.% of SS dosage, reducing the backfill cost significantly. The sintering-free process minimizes greenhouse gas emissions. It is also an attempt to explore solid waste valorization techniques and promote sustainable mining development.

**Author Contributions:** Conceptualization, B.X.; Data curation, B.X.; Formal analysis, B.X.; Funding acquisition, B.X.; Investigation, H.H.; Methodology, B.X.; Project administration, B.X.; Resources, H.H.; Software, J.Z. and H.H.; Supervision, B.X.; Validation, B.X. and H.H.; Visualization, B.X.; Writing—review & editing, B.X. All authors have read and agreed to the published version of the manuscript.

**Funding:** This research and the APA were funded by the China Postdoctoral Science Foundation (CN), Grant number 2021M690362.

**Data Availability Statement:** Data is contained within the article.

**Acknowledgments:** The authors are grateful for the help and support of Fujian Sanming Iron and Steel Group in providing the tailings and slag. The authors are grateful for the funding support by the China Postdoctoral Science Foundation.

**Conflicts of Interest:** The authors declare no conflict of interest. The funders had no role in the design of the study; in the collection, analyses, or interpretation of data; in the writing of the manuscript, or in the decision to publish the results.

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
