# Peer review of "A Green Sintering-Free Binder Material with High-Volumetric Steel Slag Dosage for Mine Backfill"

_minerals, doi:10.3390/min12081036_

Round 1
Reviewer 1 Report
The authors investigate a high-volumetric steel slag binder for sustainable cemented paste backfill. The mechanical property, hydration heat, reaction products, and pore size of the H-SSB CPB are discussed. It shows that the H-SSB has a good performance in CPB. The findings are valuable both for mine backfill and slag valorization. However, some issues need to be addressed.
i) The finding in line 206 and Fig.1 showed that the H-SSB did not prolong the dormant period and retard the setting time. The reasons should be more discussed and compared with other studies.
ii) Using steel slag as mineral admixtures in cement could have volumetric expansion risks. Can the authors give some more discussions about this in section 3.1. The following paper is recommended about the expansion properties.
http://doi.org/10.3390/min12060763
iii) In Line 272, the H-SSB CPB has 5.57% meso-pores, while the OPC-CPB are all macro-pores. Generally, what are the effects of pore size on the CPB mechanical properties?
vi) The manuscript mentioned "low-cost" in multiple places. Can the authors give some cost comparisons between the H-SSB and traditional cement?
v) The page numbers of reference 21 in Line 353 are missing.
Reviewer 2 Report
In this study, the effect of the steel-slag-based binder for CPB and explore valorization techniques of steel slag were evaluated. My comments and suggestions in this context are given below.
1. Could a numerical expression describing the conclusion of the article be given in the abstract?
2. Line 41-42: “OPC is generally partially replaced by pozzolanic materials such as metallurgical slag, fly ash, and sintered soil to reduce the cost [8, 9]”... The zeolite material, which has a pozzolanic feature and has recently been studied for its usability in CPB, can be added here. Reference: Tuylu, S. Effect of different particle size distribution of zeolite on the strength of cemented paste backfill. Int. J. Environ. Sci. Technol. 19, 131–140 (2022). https://doi.org/10.1007/s13762-021-03659-7
3. Line 43-54: “Many other researchers have developed… For example,….” In addition to these examples, the following study, which is one of the current studies, can be given as an example. “Tuylu, S. Investigation of the Effect of Using Different Fly Ash on the Mechanical Properties in Cemented Paste Backfill. J. Wuhan Univ. Technol.-Mat. Sci. Edit. 37, 620–627 (2022). https://doi.org/10.1007/s11595-022-2576-1”
4. Line 84: “Ground granulated blast furnace slag (GGBFS)” In the Introduction part, this material should be mentioned in 1 or 2 sentences.
5. Line 90: “Flue gas desulfurized gypsum (FGDG)” Similarly, it should be mentioned earlier.
6. Line 96: “Tailings” Why only the size distribution is given under the heading "Tailings". Here, the physical properties of the tailings such as specific surface area, porosity, as well as mineralogy and elemental contents, ie chemical properties, should be given.
7. Line 131: “2.3. Experimental program” was applied to which samples. Please explain briefly.
8. Line 171: “3. Results and discussion” should be rearranged according to the experimental flow given under title “2.3. Experimental program” or the experimental flow should be rearranged and given under heading 2.3.
9. Line 178-179: Other mixing results can be given in a table.
10. Line 198-199: “When a high-volumetric (50 wt.%) SS is used in the green binder (H-SSB),..” It would be better if the green binder (H-SSB) expression is shown in Table 2.
11. Line 275: “3.4. Pore structure of the CBP?? at 28-day curing”
Reviewer 3 Report
The presented study is devoted to the actual topic of searching for alternative backfill materials that can replace cement and improve its properties. The authors investigated the effectiveness of the combined binder material - finely ground steel slag, blast-furnace granulated slag, flue gas gypsum by preparing compositions with different ratios of components. The results of the study clearly have scientific and practical value.
The article undoubtedly has scientific and practical value. The subject of the article corresponds to the journal Minerals.
However, after a detailed acquaintance with the research material, I had several comments and recommendations to improve the quality of the article.
1. The authors are strongly recommended to substantiate in the article why steel slag and granulated blast-furnace slag were crushed precisely to a specific area of 430-480 m2/kg? Why is such a value of the specific area of particles adopted? Why not 200, 300 or 600 m2/kg? Please provide justification. Also indicate the method by which the specific area was determined.
2. I would recommend that the authors of the article give a comparison before declaring a significant reduction in the cost of backfill. With a classic paste backfill, when cement is used, there is no need to build a binder grinding plant. With the new type of binder recommended by the authors, the installation of mills and additional processes associated with the preparation of the binder are required. Directly on the process of grinding slag for backfilling will be significant energy costs. Please give a comparison of cement-based paste backfill and the recommended binder and prove to the reader serious savings.
3. Authors are recommended to give the chemical composition of materials for backfilling 2.1.1-2.1.4. Also give the granulometric characteristics after grinding and the physical parameters of steel and granulated blast-furnace slag, gypsum. Backfill materials form its basis, therefore, maximum initial information should be provided about them.
4. In section 2.2.1, what type of ball mill and where was the grinding carried out? Was it at Fujian Sanming Iron and Steel Group?
5. Please indicate in section 2.2.1 how many backfill samples were prepared in total? (taking into account 3 samples per mixture composition).
6. Give a short justification why in table 2 you have taken the dosage level of steel slag 35-50%? Also explain why the ratio of binder to tailings is 1:4? If necessary, provide some citations.
7. The authors in section 2.2.2 give a water-cement ratio of 1.0, and in section 2.3.3 they give a value of 0.35. Please explain why the values are different. And explain in the article more clearly.
8. In section 3.1, the authors on the graphs give the control value of the backfill strength during hardening of 7 and 28 days (in blue). Please explain these reference strength values? You do not consider an example of a specific mine in the article. I would recommend that, after the first sentence on line 195, indicate that "for comparison, data were taken from cement-based paste backfill for mine conditions ..." and quote
9. Please, for better understanding, indicate on lines 178-180 that the best recipes for strength are presented in the histograms in graph 3.
10. Please, in the description of Figure 4, give a quantitative change in the heat of hydration. What are the peak values of the studied backfill and based on cement. How many times the heat of hydration is less in the studied backfill.
11. It is not clear from the article, the authors compare the results of H-SSB with CPB. However, CPB was not investigated in the article. Probably the authors compare the results with other scientists. Please, where there is a comparison, indicate the source of literature. If it is one, at the beginning of the comparison, indicate it specifically, and then compare.
Dear authors, please consider the introduction and research material of the article, which also deals with issues related to your research topic:
Erismann, F., & Hansson, M. (2021). Efficient paste mix designs using new generation backfill admixtures–perception versus reality. In Minefill 2020-2021 (pp. 3-12). CRC Press. https://doi.org/10.1201/9781003205906-1
Petlovanyi, M., & Mamaikin, O. (2019). Assessment of an expediency of binder material mechanical activation in cemented rockfill. ARPN Journal of Engineering and Applied Sciences, 14(20), 3492-3503
Tian, X., & Fall, M. (2021). Non-isothermal evolution of mechanical properties, pore structure and self-desiccation of cemented paste backfill. Construction and Building Materials, 297, 123657. https://doi.org/10.1016/j.conbuildmat.2021.123657
Petlovanyi, M.V., Zubko, S.A., Popovych V.V., & Sai, K.S. (2020). Physicochemical mechanism of structure formation and strengthening in the backfill massif when filling underground cavities. Voprosy Khimii i Khimicheskoi Tekhnologii, (6), 142-150. https://doi.org/10.32434/0321-4095-2020-133-6-142-150
Liu, B., Gao, Y. T., Jin, A. B., & Wang, X. (2020). Influence of water loss on mechanical properties of superfine tailing–blast-furnace slag backfill. Construction and Building Materials, 246, 118482. https://doi.org/10.1016/j.conbuildmat.2020.118482
Round 2
Reviewer 2 Report
Accept in present form.
Reviewer 3 Report
I carefully reviewed the revision of the article. The authors did a good job of improving the quality of the presentation of the article material. The authors took into account recommendations for improving the study and clarified some controversial issues.
Kind regards